# Influenza virus mRNAs encode determinants for nuclear export via the cellular TREX-2 complex

Prasanna Bhat [1], Vasilisa Aksenova [2], Matthew Gazzara [3], Emily A. Rex[1], Sadaf Aslam [4,5], Christina Haddad[6], Shengyan Gao[1], Matthew Esparza[1], Tolga Cagatay[1], Kimberly Batten[1], Sara S. El Zahed [4,5], Alexei Arnaoutov [2], Hualin Zhong[7], Jerry W. Shay [1], Blanton S. Tolbert [6], Mary Dasso [2], Kristen W. Lynch[3], Adolfo García-Sastre [4,5,8,9,10] & Beatriz M. A. Fontoura [1]✉

Nuclear export of influenza A virus (IAV) mRNAs occurs through the nuclear pore complex (NPC). Using the Auxin-Induced Degron (AID) system to rapidly degrade proteins, we show that among the nucleoporins localized at the nucleoplasmic side of the NPC, TPR is the key nucleoporin required for nuclear export of influenza virus mRNAs. TPR recruits the TRanscription and EXport complex (TREX)−2 to the NPC for exporting a subset of cellular mRNAs. By degrading components of the TREX-2 complex (GANP, Germinal-center Associated Nuclear Protein; PCID2, PCI domain containing 2), we show that influenza mRNAs require the TREX-2 complex for nuclear export and replication. Furthermore, we found that cellular mRNAs whose export is dependent on GANP have a small number of exons, a high mean exon length, long 3' UTR, and low GC content. Some of these features are shared by influenza virus mRNAs. Additionally, we identified a 45 nucleotide RNA signal from influenza virus HA mRNA that is sufficient to mediate GANP-dependent mRNA export. Thus, we report a role for the TREX-2 complex in nuclear export of influenza mRNAs and identified RNA determinants associated with the TREX-2-dependent mRNA export.

Nuclear export of cellular mRNAs is mediated by intranuclear protein complexes named TREX (TRanscription and EXport) that link transcription, mRNA processing and export. TREX consists of a hexameric core termed THO (THOC1-3, 5–7), the RNA helicase UAP56 (56 kDa U2AF-associated protein), the adapter protein ALYREF (THO Complex Subunit 4), and additional proteins[1,2]. During gene expression, the TREX complex is recruited co-transcriptionally and pre-mRNA processing further mediates TREX assembly on the transcript[1]. Through the combined action of TREX complex and other mRNA export factors, the major mRNA export receptor NXF1 (nuclear RNA export

[1]Department of Cell Biology, University of Texas Southwestern Medical Center, Dallas, TX 75390, USA. [2]Division of Molecular and Cellular Biology, National Institute of Child Health and Human Development, National Institutes of Health, Bethesda, MD 20892, USA. [3]Department of Biochemistry and Biophysics, University of Pennsylvania, Philadelphia, PA 19104, USA. [4]Department of Microbiology, Icahn School of Medicine at Mount Sinai, New York, NY 10029, USA. [5]Global Health and Emerging Pathogens Institute, Icahn School of Medicine at Mount Sinai, New York, NY 10029, USA. [6]Department of Chemistry, Case Western Reserve University, Cleveland, OH 44106, USA. [7]Department of Biological Sciences, Hunter College, New York, NY 10065, USA. [8]Department of Medicine, Division of Infectious Diseases, Icahn School of Medicine at Mount Sinai, New York, NY 10029, USA. [9]Department of Pathology, Molecular and Cell-Based Medicine, Icahn School of Medicine at Mount Sinai, New York, NY 10029, USA. [10]The Tisch Cancer Institute, Icahn School of Medicine at Mount Sinai, New York, NY 10029, USA. ✉e-mail: beatriz.fontoura@utsouthwestern.edu

factor-1):NXT1 (nuclear transport factor 2-related export protein) heterodimer is recruited to the mRNA to mediate nuclear export through the NPC[1]. NXF1 can also associate with the TREX-2 protein complex (consists of (GANP, Germinal-center Associated Nuclear Protein; PCID2, PCI domain containing 2; DSS1; deleted in spilt hand/spilt foot; ENY2, Enhancer of Yellow 2 homolog; and Centrins) through its interaction with GANP[3]. The human TREX-2 component GANP is known to mediate nuclear export of a subset of cellular mRNAs[4]. During mRNA export, TREX-2 interacts with the nucleoporin TPR via GANP at the nucleoplasmic side of the nuclear pore complex (NPC)[3,5]. GANP binds the NXF1:NXT1 heterodimer[3], which then mediates translocation of the mRNP through the NPC. Once the mRNA reaches the cytoplasmic side of the NPC, the RNA-dependent ATPase and ATP-dependent RNA-unwinding activities of the DEAD-box RNA helicase DBP5 together with the mRNA export factor GLE1 induce the release of NXF1 from the mRNP, and NXF1 then recycles back to the nucleus[1].

Influenza virus mRNAs are transcribed in the host cell nucleus from eight unique viral genomic RNAs that enter the nucleus via the NPC[6]. Six mRNAs (HA, NP, NA, PB2, PB1, and PA) are intronless and hence exported without splicing[7]. Two of the viral mRNAs (M and NS) undergo alternative splicing, but a pool of unspliced messages is also exported to the cytoplasm for translation into proteins[7]. It is not yet understood how these unspliced viral mRNAs or mRNAs that undergo few splicing events are efficiently exported. Here, using the AID-tag system to rapidly degrade constituents of the NPC and the TREX-2 complex, we show that influenza virus mRNAs usurp the cellular nucleoporin TPR and the TREX-2 complex for their export from the nucleus to the cytoplasm. We find that influenza virus mRNAs contain RNA features associated with cellular mRNAs that are dependent on GANP for their nuclear export. Moreover, we identify a 45 nt RNA sequence in the 5′ end of viral HA mRNA which is sufficient to make mRNA export GANP dependent. Thus, we uncover the pathway taken by influenza virus mRNAs for nuclear export. Additionally, we identify RNA features and a signal that mediate GANP-dependent mRNA export.

## Results

### The nucleoporin TPR (but not NUP153 and NUP50) is required for influenza A virus mRNA nuclear export

During mRNP export through the NPC, the first structure mRNPs encounter is the basket localized at the nucleoplasmic side of the NPC. Thus, we sought to determine the nuclear basket nucleoporins (Nups) that are required for viral mRNA export. We used the AID-tag system to selectively and rapidly degrade individual nuclear basket Nups in DLD-1 cells, as we previously reported[5], to prevent long term effects of depletion. DLD-1 cells expressing AID-tagged NUP153, NUP50, or TPR (NUP153[AID], NUP50[AID], [AID]TPR cells)[5] were infected with IAV (A/WSN/33). After 2 h 45 min of infection, auxin was added to degrade the specific Nup and infection was maintained for an additional 5 h 15 min (Fig. 1a). Cells were then subjected to single-molecule RNA fluorescence in situ hybridization (smRNA FISH) to detect and quantify the viral M, NS, and HA mRNAs, which are differentially processed. The M and NS mRNAs undergo alternative splicing whereas the HA mRNA is not spliced[7]. The probes used for M and NS mRNAs can recognize both unspliced and spliced forms[8]. We found that depletion of NUP153 or NUP50 did not alter nuclear export of these viral mRNAs (Fig. 1b–o). These results were further corroborated by quantification of fluorescence intensity in the whole cell and in the nucleus, and the results are presented as % nuclear mRNA (Fig. 1e–g, l–n). The levels of NUP153 and NUP50 proteins, before and after auxin treatment, were detected by western blot and auxin treatment led to efficient degradation of both proteins (Fig. 1h, o). In contrast, depletion of TPR robustly inhibited nuclear export of the M, NS and HA mRNAs, as shown by smRNA FISH (Fig. 1p–r) and quantification of fluorescence intensity (Fig. 1s–u). The results of auxin treated [AID]TPR cells were compared to parental DLD-1

cells as levels of TPR were slightly decreased in untreated [AID]TPR cells (Fig. 1v) and were sufficient to induce mRNA export block (Fig. 1p–u). Similar results were obtained by subcellular fractionation in which additional influenza virus mRNAs were quantified in nuclear and cytoplasmic fractions and were shown to be significantly retained in the nucleus upon TPR degradation (Supplementary Fig. 1a). Thus, among basket nucleoporins TPR is critical for influenza virus mRNA export.

### Nuclear export of influenza virus mRNAs is dependent on constituents of the TREX-2 complex

Since TPR is required for IAV mRNA export (Fig. 1) and recruits the TREX-2 complex to the NPC[5] (Fig. 2a), we tested whether constituents of this complex are involved in exporting viral mRNAs. We carried out experiments in DLD-1 cells expressing AID-tagged GANP or PCID2 ([AID]GANP, [AID]PCID2)[5], which are key constituents of the TREX-2 complex. The experimental conditions and assays were performed as described in Fig. 1. We found that depletion of GANP (Fig. 2b–g) or PCID2 (Fig. 2i–n) by auxin treatment prevented nuclear export of viral M, NS, and HA mRNAs compared to cells not treated with auxin. The [AID]GANP cells untreated with auxin showed similar distribution of viral mRNAs as parental DLD1 cells (Fig. 1p–u). As shown by western blot, GANP and PCID2 proteins were effectively degraded by auxin treatment (Fig. 2h, o). Additionally, the viral NP (nucleoprotein) and NA (neuraminidase) mRNAs also depend on GANP and PCID2 for export (Supplementary Figs. 1b, c and 2). These results are further corroborated by subcellular fractionation in which additional viral mRNAs were significantly retained in the nucleus upon depletion of GANP or PCID2 (Supplementary Fig. 1b, c). The nuclear export inhibition of viral mRNA was also observed in A549 cells and using diverse viral strains (A/California/07/2009; A/Wyoming/03/2003; A/Vietnam/1203/2004) (Supplementary Fig. 3). We have also observed a decrease in viral protein levels and viral replication upon GANP knockdown (Fig. 3). Thus, these findings indicate that IAV exploits the TREX-2 complex for nuclear export of its mRNAs. Furthermore, using AID-tagged NXF1 cells, we confirmed that these viral mRNAs require NXF1 for their export (Supplementary Fig. 4), which is likely recruited by the TREX-2 complex via GANP. The latter results are consistent with previous findings using siRNAs in which NXF1 was shown to be required for influenza virus mRNA export[9–12].

### Influenza A virus mRNAs contain RNA features associated with cellular mRNAs that are dependent on GANP for their nuclear export

We then sought to define the determinants in the viral mRNAs that dictate their dependence on TREX-2 for nuclear export. Since TREX-2 appears to mediate nuclear export of a subset of cellular mRNAs[4] and viral mRNAs are usurping this pathway (Fig. 2), we first carried out a systematic identification of the cellular mRNAs whose export is dependent on TREX-2. As GANP is a key constituent of the TREX-2 complex, we depleted GANP to identify cellular mRNAs that are dependent on TREX-2 for nuclear export. [AID]GANP cells were treated with auxin for 8 h to degrade GANP and then fractionated into nuclear and cytoplasmic fractions (Fig. 4a). RNA was purified from whole cell lysates, nuclear and cytoplasmic fractions, and each sample was subjected to RNA-seq analysis (Fig. 4a and Supplementary Data 1). The presence of high levels of MALAT1 RNA in the nuclear fractions and higher levels of GAPDH or actin mRNAs in the cytoplasm than in the nucleus validated the fractionation procedure (Supplementary Data 1 and Supplementary Fig. 5).

For selecting cellular mRNAs dependent on GANP for nuclear export, we first identified the mRNAs whose levels were not significantly altered ($1 < \log_2$ fold change $> -1$) in the total cell lysate of GANP-depleted cells compared to control cells. The rationale for this selection is that changes in total mRNA levels may involve other RNA

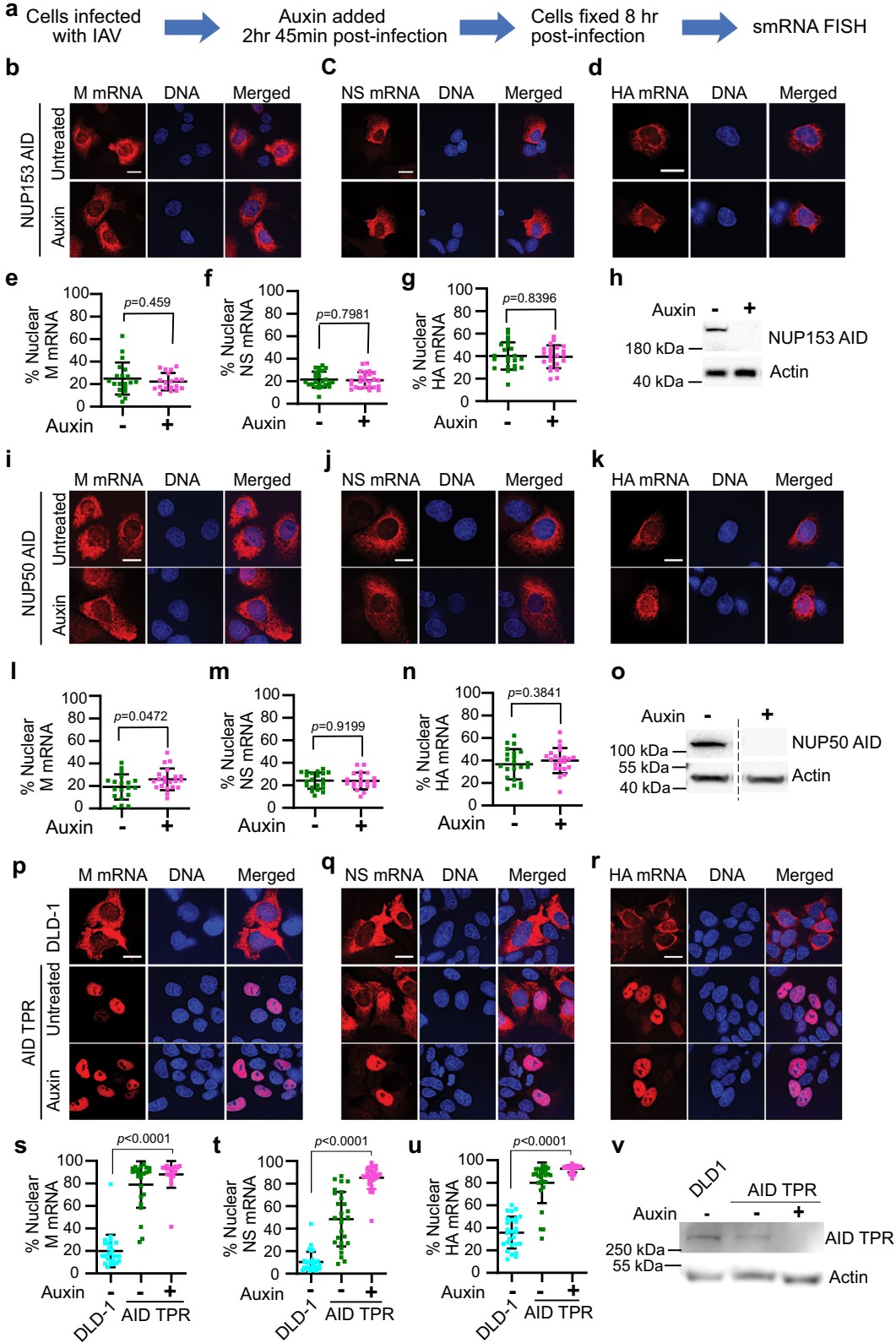

processing pathways. Thus, from this group of mRNAs that are not altered at the whole cell level, the mRNAs whose relative change in nuclear to cytoplasmic (N/C) ratio was >3.5 upon GANP depletion were considered as export inhibited, and mRNAs with relative change in N/C ratios between 1.5 and 0.66 were considered as not affected by GANP depletion. In this manner, we selected mRNAs whose nuclear export is inhibited with no significant change in their total cellular levels. Upon GANP degradation, we found that 398 mRNAs were accumulated in the

nucleus without significant changes in their total levels indicating their dependence on GANP for nuclear export (Supplementary Data 1). Nuclear export of 3,694 mRNAs was not affected upon GANP depletion (Supplementary Data 1).

Analysis of shared features among the mRNAs that are GANP-dependent for nuclear export showed that this subset of mRNAs has a small number of exons and long exons compared to mRNAs whose localization was not affected by GANP depletion (Fig. 4b–d).

**Fig. 1 | The nucleoporin TPR (but not NUP153 and NUP50) is required for influenza A virus mRNA nuclear export. a** DLD-1 cells expressing endogenously AID-tagged basket nucleoporins (NUP153^AID, NUP50^AID and ^AIDTPR cells) were infected with influenza A virus (A/WSN/33) and after 2 h 45 min cells were untreated or treated with auxin to degrade AID-tagged proteins. After 8 h of infection, cells were fixed and subjected to smRNA FISH to detect viral M, NS, and HA mRNAs with probes labeled with Quasar 570. DNA was stained with Hoechst. **b–d** NUP153^AID cells were infected and treated with auxin as described in **a**. Viral mRNAs and DNA were detected as in (**a**). Scale bar, 10 μm. Data are representative of three independent experiments. **e–g** Fluorescence intensity of each viral mRNA was quantified in the whole cell and in the nucleus using Imaris software (Bitplane). Percentage of nuclear values for each viral mRNA are shown for each cell (dot) (**e** Untreated *n* = 20 cells and auxin treated *n* = 19 cells, **f** Untreated *n* = 24 cells and auxin treated *n* = 25 cells, **g** Untreated *n* = 22 cells and auxin treated *n* = 22 cells). For each condition, cells used for quantification were derived from three independent experiments. **h** Western blot was performed using anti-NUP153 antibody to confirm NUP153 degradation. **i–k** NUP50^AID cells were infected and treated with auxin as described in (**a**). Viral mRNAs and DNA were

detected as in **a**. Scale bar, 10 μm. Data are representative of three independent experiments. **l–n** Quantification of fluorescence intensity was performed as in **e** (**l** Untreated *n* = 19 cells and auxin treated *n* = 21 cells, **m** Untreated *n* = 21 cells and treated auxin *n* = 20 cells, **n** Untreated *n* = 22 cells and auxin treated *n* = 22 cells). For each condition, cells used for quantification were derived from three independent experiments. **o** Western blot was performed using anti-NUP50 antibody to confirm NUP50 degradation. **p–r** DLD-1 and ^AIDTPR cells were infected and untreated or treated with auxin as described in **a**. Viral mRNAs and DNA were detected as in (**a**). Scale bar, 10 μm. Data are representative of three independent experiments. **s–u** Quantification of fluorescence intensity was performed as in **e** (**s** DLD-1 *n* = 26 cells, untreated *n* = 24 cells and auxin treated *n* = 20 cells, **t** DLD-1 *n* = 28 cells, untreated *n* = 27 cells and auxin treated *n* = 27 cells, **u** DLD-1 *n* = 27 cells, untreated *n* = 27 cells and auxin treated *n* = 26 cells). For each condition, cells used for quantification were derived from three independent experiments. **v** Western blot was performed using anti-TPR antibody to confirm TPR degradation. Graphs show mean ± SD. *p* values were calculated using unpaired two tailed Student's t test (GraphPad Prism 9). *p* values ≤0.05 are considered significant.

Additionally, the mRNAs dependent on GANP for export have longer 3' UTR (Fig. 4e) and a slightly overall lower GC content (Fig. 4f), including lower GC content at specific regions in the 5' end of mRNAs (Supplementary Fig. 6), than the mRNA population not affected by GANP degradation. In contrast, the pre-mRNA length is not different between these two groups of mRNAs (Fig. 4g). The 5' UTR and the mRNA coding region of GANP-dependent mRNAs are only slightly longer than the mRNAs whose export is not affected by GANP depletion (Supplementary Fig. 7). Selected mRNAs accumulated in the nucleus by GANP depletion were further validated using qPCR (Supplementary Fig. 8). These mRNAs again showed higher nuclear to cytoplasmic ratios (N/C) upon GANP depletion compared to N/C ratios in the presence of GANP, corroborating the RNA-seq results. Altogether, these results suggest a potential combination of RNA features that may determine GANP/TREX2-dependency for nuclear export. We then compared the mRNA features of the influenza virus mRNAs with the top 10 cellular mRNAs that are dependent on GANP for export and the 10 top cellular mRNAs that are independent on GANP (Fig. 5a, b). We found that the small number of exons and the high mean exon length are the common determinants between the influenza virus mRNAs and cellular mRNAs that are dependent on GANP for nuclear export (Fig. 5). Together, these results suggest that small exon number and high mean exon length are features that may contribute to GANP-dependent export of cellular and influenza A virus mRNAs.

## A sequence at the 5' end of influenza virus HA mRNA coding region can determine GANP-dependent mRNA export

Next, we selected the viral HA mRNA as a model to further investigate GANP dependency. We first used a recombinant influenza virus expressing eGFP mRNA in which the eGFP sequence is flanked by the 5' end of the HA mRNA [nt 1–77 (5' UTR: nt 1–32 + coding region: nt 33–77)] and the 3' end [nt 1651–1754 (last 80 nt of the HA mRNA coding region: nt 1651–1730 + 3'UTR: nt 1731–1754)] (Fig. 6a). ^AIDGANP cells were infected with this recombinant virus and auxin was added 2 h 45 min post-infection to degrade GANP. After 5 h 15 min of auxin treatment, cells were subjected to smRNA FISH to detect the eGFP mRNA (Fig. 6b). We found that eGFP mRNA flanked by the 5' and 3' ends of the viral HA mRNA was retained in the nucleus upon GANP degradation (Fig. 6b) and quantification of fluorescence intensity in the whole cell and in the nucleus further demonstrated the percentage increase in nuclear retention upon GANP depletion (Fig. 6c). These results indicated that there is a determinant(s) within the 5' end and/or the 3' end of the viral HA mRNA that makes mRNA export dependent on GANP.

We then expressed eGFP mRNA alone from a mammalian expression vector (pCAG) in ^AIDGANP cells. After 20 h of transfection, cells were treated with auxin for 5 h 15 min to deplete GANP. In this

context, nuclear export of eGFP mRNA was only slightly dependent on GANP (Fig. 7a). In the context of the pCAG vector, the eGFP mRNA undergoes one splicing event and has a long exon, which are features common to the cellular mRNAs that are dependent on GANP for export (Fig. 4). However, eGFP mRNA also has a high GC content (61.5%), which is a feature known to promote mRNA export[13] that may significantly counteract GANP requirement and explain the low GANP dependency of eGFP mRNA. Using this same mammalian expression vector, we generated a series of mutants in the HA sequences (Fig. 7a). When eGFP mRNA was flanked by the 5' end and 3' end of the viral HA mRNA, this mRNA showed increased nuclear retention compared to the eGFP mRNA alone, suggesting that the 5' end and/or the 3' end of the HA mRNA contain determinants for GANP-mediated nuclear export. We then fused the 5' end of the HA mRNA (nt 1-77) alone with eGFP and expressed it in the presence of GANP or upon GANP depletion. We found that this 5' end region of the HA mRNA is sufficient to mediate GANP-dependent export, as eGFP mRNA was retained in the nucleus upon GANP degradation. In contrast, nuclear export of eGFP mRNA fused with the 3' end of the HA mRNA did not alter eGFP mRNA upon GANP depletion. These results indicated that the 5' end of the HA mRNA alone has determinants to dictate nuclear export via GANP. To further map this region, we fused the either 5' UTR of the HA mRNA (nt 1-32) or the first 45 nt from 5' end of HA mRNA coding region to eGFP and found that the 5'UTR does not alter GANP dependency of eGFP mRNA upon GANP depletion while the first 45 nt from the HA mRNA coding region was able to significantly increase GANP dependency of eGFP mRNA (Fig. 7a). These results indicated that the first 45 nt of the HA mRNA coding region (nt 33-77) are important to make the 5'end of the HA mRNA a determinant/signal for nuclear export via GANP. Of note, this 45 nt sequence has a region that is GU-rich (Fig. 7b), a sequence motif that serves as a recognition sequence for several RBPs[14], and has two predicted stem-loop structures (Fig. 7c).

The 45-nucleotide HA mRNA sequence was aligned against the BV-BRC database and the NCBI database to record sequence conservation. Similar sequence alignment results were obtained in both databases. The 45-nucleotide region of HA mRNAs was shown to be conserved across several influenza A strains. The first 13 strains depicted in Supplementary Fig. 9a are shown to be identical, while strains numbered 13 through 22 have shown to hold at least a 91% identity match. In addition to sequence conservation, secondary structure predictions were compared between these 22 strains using the *RNAstructure* webserver. The secondary structures of the different sequences obtained from the sequence alignment were predicted and grouped in Supplementary Fig. 9b according to their sequence similarities. The first secondary structure corresponding to strains 1-13, which are identical to the 45-nucleotide HA mRNA region, has shown to form two distinct stem-loops. The first demonstrating a low probability of

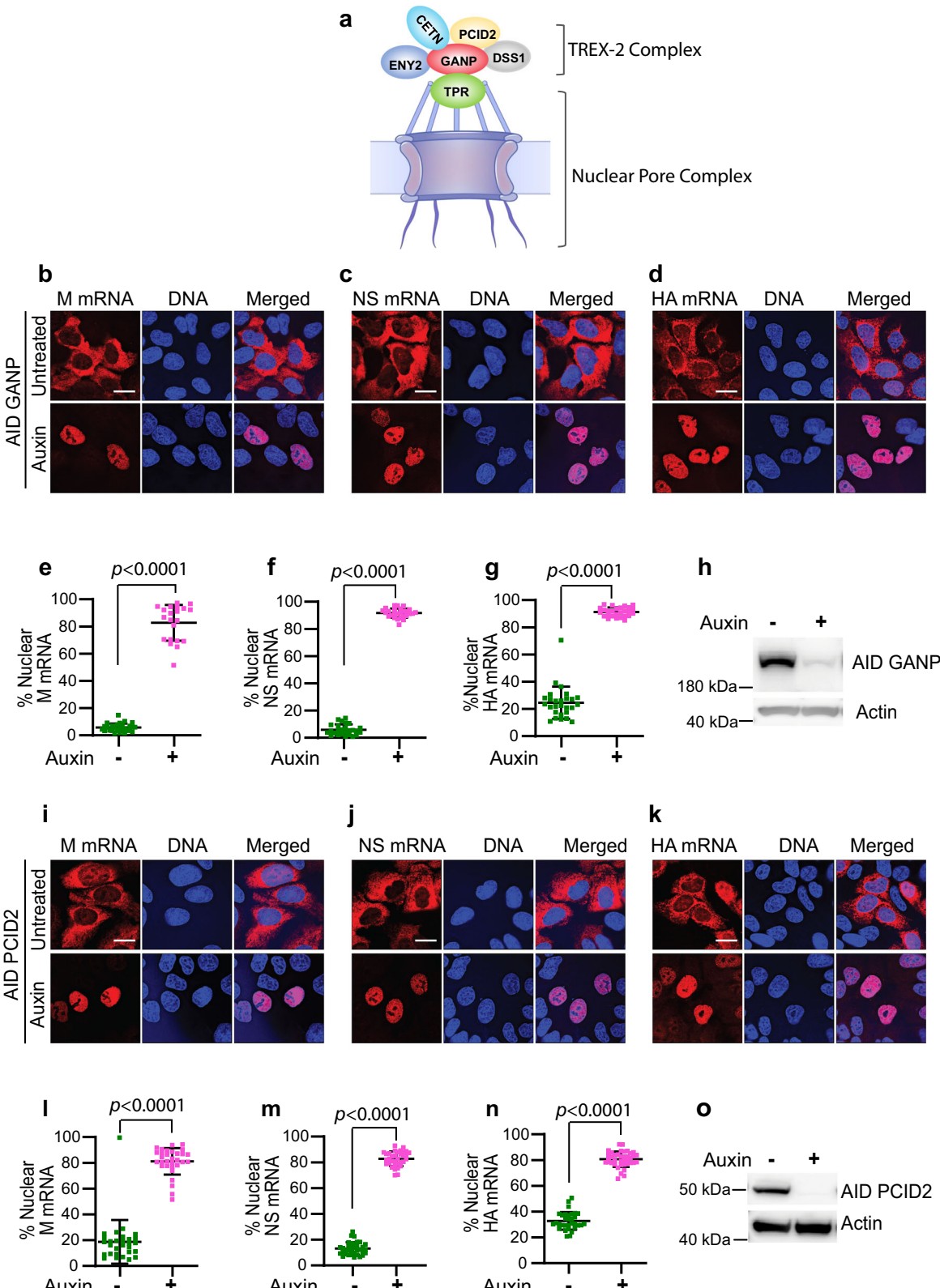

folding of around 50%, while the second stem-loop shows a higher probability of 80%. In strain 14, the single nucleotide change caused the first stem-loop to unfold, but maintains the formation of the second stem-loop, which is identical to the original structure. In strains 15 through 20, a three-nucleotide alteration caused a complete alteration in the predicted secondary structure compared to the HA mRNA region. Similar observations can be made for strains 21 and 22 with a

folding probability of 50%. These low probabilities can be attributed to their dependence on other stem-loops in close proximity to stabilize their structures.

We have also observed that when the 5′ and 3′ ends of the HA mRNA flanking eGFP was expressed in the context of the viral mini-genome system, in which the mRNA is synthesized by the viral polymerase, the GANP-dependency for export was enhanced as compared

**Fig. 2 | Nuclear export of influenza virus mRNAs is dependent on constituents of the TREX-2 complex. a** Schematic representation of the TREX-2 complex interaction with the nucleoporin TPR. **b**−**d** [AID]GANP cells were infected with influenza A virus (A/WSN/33) and 2 h 45 min post-infection cells were untreated or treated with auxin to degrade GANP. After 8 h of infection, cells were fixed and subjected to smRNA FISH to detect M, NS and HA mRNAs with probes labeled with Quasar 570. DNA was stained with Hoechst. Scale bar, 10 μm. Data are representative of three independent experiments. **e**−**g** Fluorescence intensity of each viral mRNA was quantified in the whole cell and in the nucleus using Imaris software (Bitplane). Percentage of nuclear values for each viral mRNA are shown for each cell (dot) (**e** Untreated *n* = 28 cells and auxin treated *n* = 21 cells, **f** Untreated *n* = 26 cells and auxin treated *n* = 27 cells, **g** Untreated *n* = 27 cells and auxin treated *n* = 27 cells). For each condition, cells used for quantification were derived from three independent experiments. **h** Western blot was performed using anti-HA tag antibody to confirm GANP degradation. **i**−**k** [AID]PCID2 cells were infected and untreated or treated with auxin as in (**b**). Viral mRNAs and DNA were detected as in **b**. Scale bar, 10 μm. Data are representative of three independent experiments. **l**−**n** Quantification of fluorescence intensity was performed as in **e** (**l** Untreated *n* = 29 cells and auxin treated *n* = 31 cells, **m** Untreated *n* = 35 cells and auxin treated *n* = 34 cells, **n** Untreated *n* = 32 cells and auxin treated *n* = 39 cells). For each condition, cells used for quantification were derived from three independent experiments. **o** Western blot was performed using anti-HA tag antibody to confirm PCID2 degradation. Graphs show mean ± SD. *p* values were calculated using unpaired two tailed Student's t test (GraphPad Prism 9). *p* values ≤ 0.05 are considered significant.

to the expression of the same mRNA in the context of plasmid. These results suggest that the viral polymerase may have a role in facilitating GANP-mediated export. In summary, we identified a sequence in the 5' end of the influenza virus HA mRNA that can mediate mRNA export through the TREX-2 complex.

## Discussion

Here, we report key players involved in nuclear export of influenza virus mRNAs. We found that the viral mRNAs usurp the TREX-2 mRNA export complex and the nucleoporin TPR to mediate their export from the nucleus. The requirement of TPR for viral mRNA export and not of the other basket Nups (NUP153 and NUP50) is consistent with our previous results showing that TPR, and not NUP153 and NUP50, is required for GANP (TREX-2 component) localization to the NPC[5]. Hence, depletion of TPR likely inhibited viral mRNA export because of the decrease in GANP localization to the NPC. Depletion of TREX-2 components or TPR resulted in a robust nuclear export inhibition of influenza virus mRNAs, demonstrating an important and previously unknown role for the TREX-2 complex in the influenza virus life cycle. All viral mRNAs were tested and are dependent on the TREX-2 complex components for export. These mRNAs include the ones that undergo splicing, M and NS, and the others that are not spliced. In addition to the TREX-2 complex, IAV M mRNA export also requires ALY/REF, UAP56[8], NXF1[9–12] (Supplementary Fig. 4). Therefore, the TREX-2 complex likely functions together with constituents of the TREX complex to mediate influenza virus mRNA export. It has been shown that depletion of TPR significantly reduced the probability of mRNAs entering the NPC for export[15]. However, NXF1 localization and dynamics at the NPC is independent of TPR[5]. Since GANP interacts with NXF1[3] and TPR[4], it is possible that GANP interaction with TPR may increase association of GANP-bound mRNPs with the NPC and GANP also acts as an adapter to recruit NXF1.

We systematically identified cellular mRNAs that are dependent on GANP for export to gain insights into shared RNA features between the cellular and viral mRNAs that could drive mRNA export via GANP. From RNA-seq data analysis we found that low exon number, longer exons, longer 3' UTR, and low GC content are associated with GANP-dependent export. Like GANP-dependent cellular mRNAs, IAV mRNAs have low exon numbers (1 or 2 exons) and high mean exon length indicating that these shared RNA features may contribute to the export of influenza virus mRNAs via GANP. Previously, a subset of cellular mRNAs whose export is dependent on GANP has been identified using siRNA[4]. Here we used the AID-tag system which allows quick degradation of GANP, avoiding long-term effects of depletion. Furthermore, earlier work utilized microarray analysis of the cytoplasmic fraction alone to identify the GANP-dependent mRNAs[4], while here we have performed RNA-Seq of both nuclear and cytoplasmic fractions. In the previous study, mRNAs with low exon number were reported to be associated with GANP dependency[4], which is in agreement with our findings. In our current study, we report new additional RNA features (long exons, longer UTRs, and low GC content) associated with GANP

dependency. In another study, knockdown of TPR was shown to inhibit export of intronless mRNAs or mRNAs with low exon numbers[16]. These results further corroborate our findings as TPR is required for GANP localization at the NPC[5].

Splicing promotes mRNA export by recruiting NXF1 through the exon junction complex (EJC) and/or SR proteins[17]. One would expect that export of unspliced mRNAs and of mRNAs that undergo few splicing events like influenza virus mRNAs, would be associated with a fewer EJCs and/or SR proteins, which would recruit less NXF1 than mRNAs with many exons. Thus, the unspliced mRNAs or the mRNAs with few exons may use GANP to recruit NXF1. In addition, GANP-mediated docking of mRNPs to the NPC via TPR may promote export of mRNAs associated with less NXF1 molecules. In contrast, mRNAs with high exon numbers may be decorated with a large number of mRNA export factors (such as ALY/REF and NXF1)[17], making them less dependent or independent on GANP for nuclear export. It has been previously shown that nuclear export of mRNAs with high mean exon length[18] and long 3' UTR[19] are more sensitive to NXF1 knockdown than mRNAs with a small mean exon length and 3' UTR. These results suggest that this class of mRNAs recruits NXF1 less efficiently or requires higher number of NXF1 molecules to get exported. This NXF1 phenotype together with the RNA features of this subset of mRNAs may again indicate that these mRNAs interact with GANP to efficiently recruit NXF1 and dock at the NPC. Together, these findings are consistent with the RNA features we identified in both cellular and viral mRNAs that are dependent on GANP for nuclear export.

Additionally, it is known that mRNAs with high GC content at the 5' end of cellular mRNAs are more efficiently exported from the nucleus than mRNAs with low GC content[13], which may need additional factors, such as GANP, to facilitate export. Recently, the RBM33 protein was shown to bind to high GC stretches on mRNAs and recruit NXF1 for their export[20]. We found that the GANP-dependent mRNAs have a lower overall GC content than the GANP-independent mRNAs including regions within the first 75 nt at the 5' end, therefore it is possible that GANP-independent mRNAs may use small GC-rich regions to recruit mRNA export factors to mediate export independent or less dependent on GANP. We have then further analyzed the viral HA mRNA and found that the 5' end (1-77nt: 5'UTR + 5' end of the coding region) contains sufficient information required to make a reporter GFP mRNA GANP-dependent for export. The HA 5' UTR fused with the eGFP mRNA failed to increase GANP dependency while the first 45 nt of the HA coding region were sufficient to promote export through GANP. This 45 nt region is conserved among influenza A viruses and is predicted to potentially form stem-loop structures (Supplementary Fig. 9). Thus, these results indicate that the first 45 nt of the HA mRNA coding region have critical determinants for mRNA nuclear export via GANP and may interact with specific RNA binding proteins to mediate GANP-dependent export, which is a topic of future research. We have also observed that the viral polymerase further enhanced GANP-dependency when the eGFP mRNA flanked by the 5' and 3' ends of

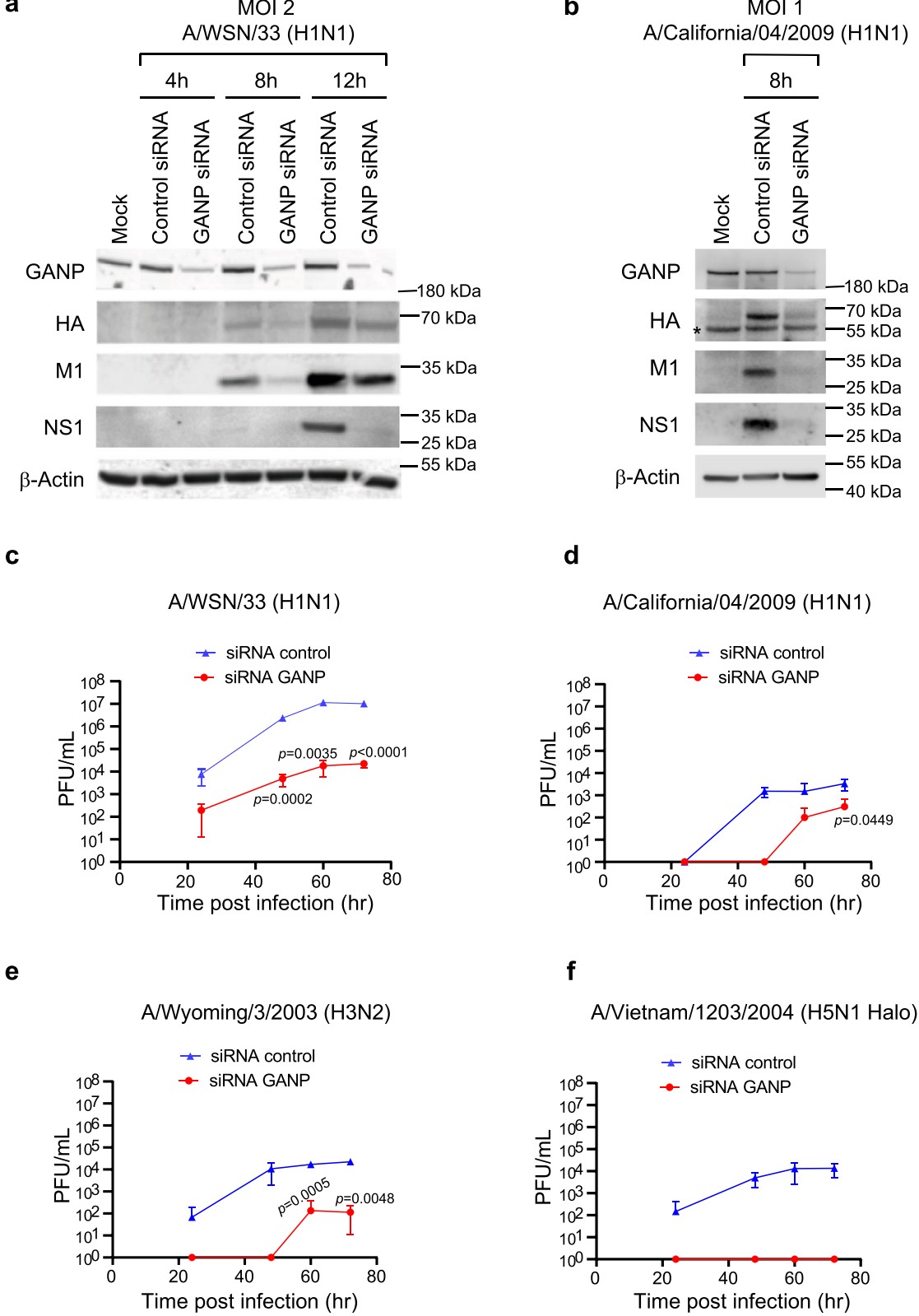

**Fig. 3 | Low levels of GANP decrease influenza virus protein levels and inhibit influenza virus replication. a, b** A549 cells transfected with siRNA control or siRNA targeting GANP were infected with A/WSN33 at MOI 2 for 4, 8, or 12 h (**a**) or with A/California/04/2009 at MOI 1 for 8 h (**b**). Cell lysates were subjected to western blot analysis to detect the depicted viral proteins (HA, NS1, and M1). β-Actin serves as a loading control. Asterisk marks a cross-reacting band. **c–f** Viral replication was measured in A549 cells transfected with siControl or siGANP siRNAs and infected with the influenza viruses A/WSN/33 (**c**), A/California/04/2009 (**d**), A/Wyoming/3/2003 (**e**), and A/Vietnam/1203/2004 (**f**) at MOI 0.01. Supernatants were collected at 24, 48, 60, and 72 h post infection and viral titers (PFU/ml) were quantified by plaque assays. Graphs represent three independent experiments and show mean +/− SD. $p$ values were calculated using unpaired two tailed Student's $t$ test (GraphPad Prism 9). $p$ values ≤ 0.05 are considered significant.

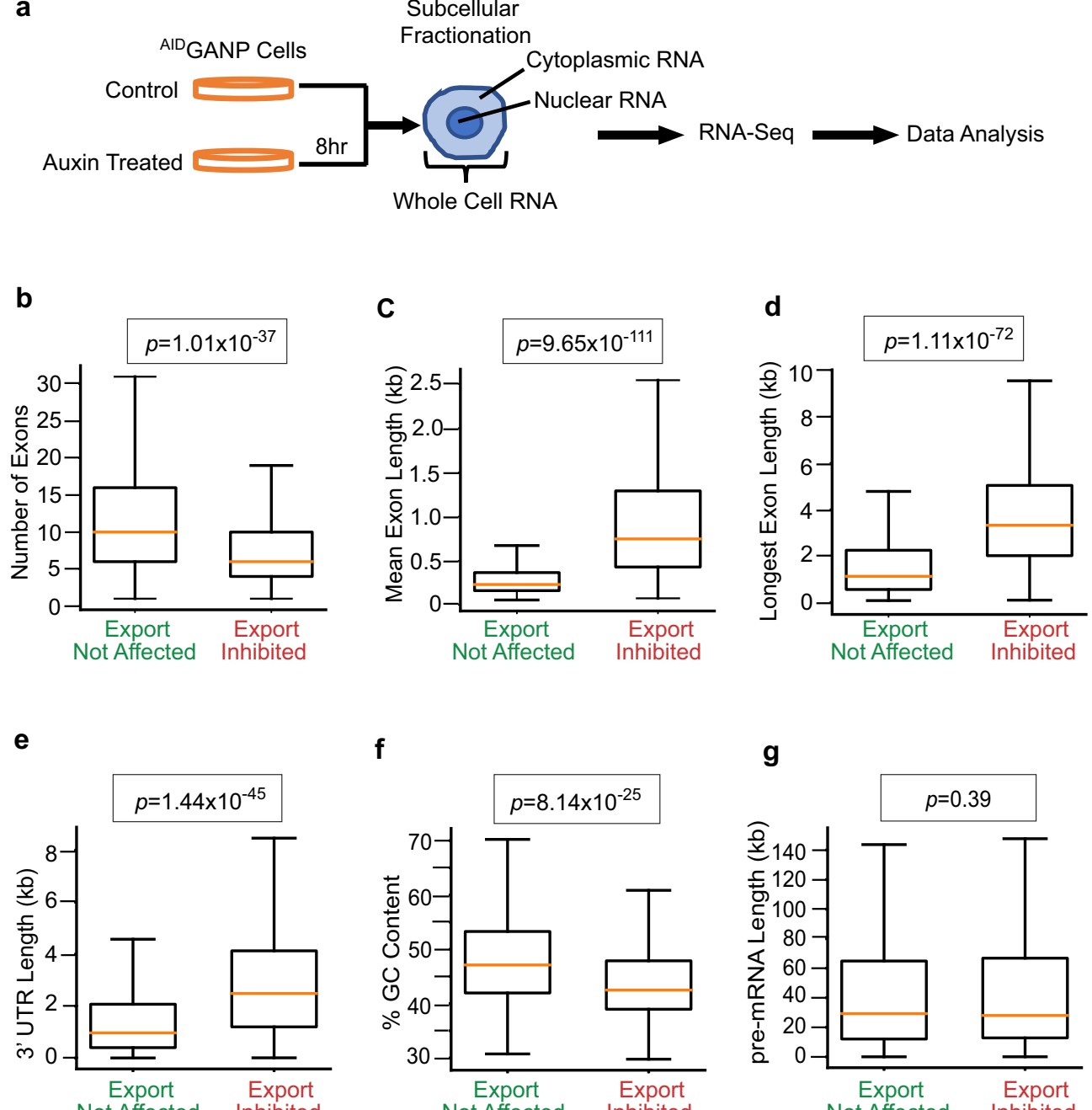

**Fig. 4 | RNA features associated with cellular mRNAs that are dependent on GANP for their nuclear export. a** [AID]GANP cells were untreated or treated with auxin to degrade the GANP protein. After 8 h, RNA was isolated from whole cells or cells were subjected to subcellular fractionation to isolate RNA from cytoplasmic and nuclear fractions. RNA-seq was performed with these RNA samples and data were analyzed to identify RNA features associated with cellular mRNAs that are dependent on GANP for their nuclear export. **b–g** Boxplots show the distribution of the RNA features between export inhibited (right) and export not affected (left) mRNAs upon GANP degradation. The box represents the 75th and 25th percentiles of the distribution, the orange line indicates the median, and whiskers indicate 1.5 times the interquartile range of the data for each group. Mann–Whitney $U$-test (two-tailed test) was used to compare export inhibited (398 genes) and export not affected (3694 genes). Data are derived from two independent RNA-Seq experiments.

the HA mRNA was expressed in the context of the minigenome system. This finding may indicate that the viral polymerase may facilitate recruitment of factor(s) involved in the TREX-2 export pathway. Together, these findings suggest that influenza A virus mRNAs as well as a subset of cellular mRNAs encode a combination of RNA features that determine GANP-dependent nuclear export. In sum, we uncovered a novel role for the TREX-2 complex in nuclear export of influenza virus mRNAs and identified RNA determinants, which alone or in combination, are associated with GANP dependency for nuclear

export. This knowledge provides insights into potential antiviral strategies that could be developed for inhibiting influenza virus mRNA export and consequently virus replication.

## Methods

### Cell lines and cell culture

DLD-1, NUP50[AID], NUP153[AID], [AID]TPR, [AID]GANP, and [AID]NXF1 cells[5] were generated as previously described[5]. To generate the [AID]PCID2 cells, the *PCID2* gene was tagged with a 1xmicroAID[21] and HA-tag at the

**a**

| Gene Name | Number of Exons | Mean Exon Length (nt) |
|---|---|---|
| **GANP Dependent mRNA Export** | | |
| *VGLL3* | 4 | 2609.50 |
| *RAB33B* | 2 | 2124.00 |
| *LUZP1* | 4 | 2105.00 |
| *REST* | 4 | 1827.00 |
| *ZNF107* | 4 | 1416.00 |
| *TRIM59* | 3 | 1274.66 |
| *ZNF12* | 5 | 1015.00 |
| *FAM8A1* | 5 | 945.00 |
| *ZNF92* | 4 | 791.25 |
| *METTL18* | 2 | 731.00 |
| | | |
| **GANP Independent mRNA Export** | | |
| *DOCK8* | 48 | 155.16 |
| *CYFIP1* | 31 | 220.19 |
| *HADHA* | 20 | 147.15 |
| *DNAJC11* | 16 | 200.06 |
| *HADHB* | 16 | 124.81 |
| *NBPF3* | 15 | 251.13 |
| *AGBL5* | 15 | 212.00 |
| *GNB1* | 12 | 263.58 |
| *AASDHPPT* | 6 | 461.00 |
| *VAMP3* | 5 | 435.00 |

**b**

| IAV mRNAs | Number of Exons | Mean Exon Length (nt) |
|---|---|---|
| *PB2* | 1 | 2341 |
| *PB1* | 1 | 2341 |
| *PA* | 1 | 2233 |
| *HA* | 1 | 1778 |
| *NP* | 1 | 1565 |
| *NA* | 1 | 1413 |
| *M* | 1/2 | 1027/169.5 |
| *NS* | 1/2 | 890/209 |

**Fig. 5 | IAV mRNAs contain RNA features associated with cellular mRNAs that are dependent on GANP for their nuclear export. a** Number of exons and mean exon length are shown for the top mRNAs from the GANP dependent and independent mRNA groups. **b** Number of exons and mean exon length are shown for influenza A virus (A/WSN/33) mRNAs.

N-terminus using the following gRNAs CATGGGAGCGCCGCCGAA and GCTGCAGGTACTGGTTAA (Supplementary Fig. 10a). The repair template for PCID2 gene targeting has been assembled using Gibson assembly reaction (E2621S, NEB) and Platinum SuperFi DNA polymerase (12351010) with the oligonucleotides listed in Supplementary Table 1. To verify the homozygosity of clones, we performed genotyping of DLD-1 cells using the protocol previously described[5] and two independent sets of oligonucleotides: F*in* CCT CGCCACTGGACCC, R*in* CCACAGCCTCGCGCT, F*out* CCGGGACGT GCCCTAAC, R*out* CCTCAGCCCACCCATTCTTATT (Supplementary Fig. 10b). Ubiquitin ligase TIR1 was knocked-in into the Regulator of Chromosome Condensation 1 (RCC1, NC_000001.11) locus using CRISPR/Cas9-mediated recombination as described previously[5] (Supplementary Fig. 10a). AID-mediated degradation of PCID2 in the presence of auxin was confirmed by Western blot analysis (Supplementary Fig. 10c) and by label-free quantification proteomics of cell lysates (Supplementary Dataset 2). Total cell lysates were prepared according to a previously published protocol[5]. Auxin (1 mM) was added to degrade the specific AID-tagged proteins, as described in

the figure legends. MDCK cells were used for virus amplification. Human lung adenocarcinoma epithelial cells (A549) and MDCK cells were obtained from ATCC (American Type Culture Collection). All cells were cultured in high-glucose DMEM (Gibco) medium with 10% FBS (Atlas) and Pen/Strep antibiotics. Cells were cultured at 37 °C in the presence of 5% $CO_2$.

**Antibodies**

The following antibodies were used for western blot analysis: NUP50 (Catalog # A301-782A; dilution 1:500), NUP153 (Catalog # A301-789A; dilution 1:500) from Bethyl Laboratories, HA-tag(C29F4) (Catalog # 3724; dilution 1:750) from Cell Signaling Technology, β-actin (13E5) (Catalog # 4970 S; dilution 1:1500) from Cell Signaling Technology, GANP (Catalog # A303-127A; dilution 1:500) from Bethyl Laboratories, NS1 (Catalog # GTX125900; dilution 1:500) from GeneTex, anti-influenza A virions (Catalog # B65141G; dilution 1:1000) from Meridian Bioscience, β-actin (Catalog #A1978; dilution 1:5000) from Sigma Aldrich, and TPR rabbit polyclonal antibody (dilution 1:200) as previously described[22].

**a**

Hemagglutinin-pseudotyped
eGFP-expressing IAV Virus

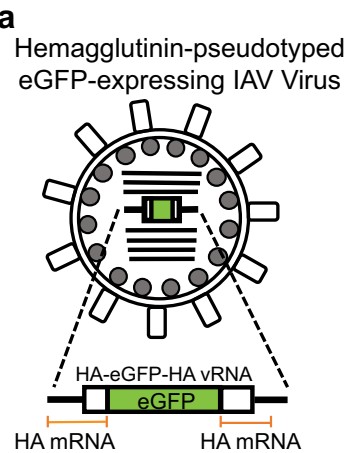

HA-eGFP-HA vRNA

HA mRNA          HA mRNA
5' end (77nt)      3' end (104nt)

**b**

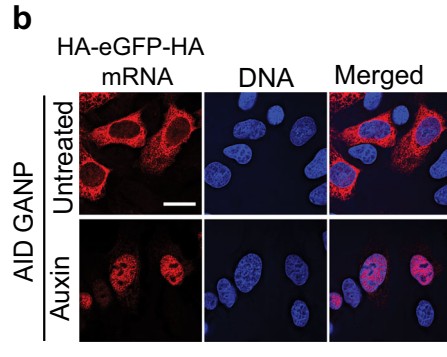

**c**

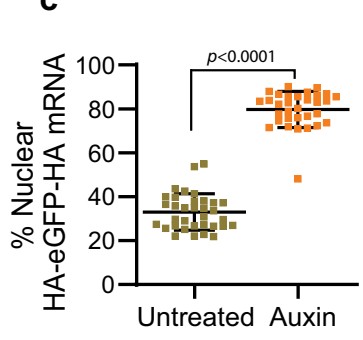

Fig. 6 | **eGFP mRNA flanked by the 5' and 3' ends of the HA mRNA is dependent on GANP for nuclear export. a** Schematic representation of influenza virus expressing hemagglutinin (HA)-pseudotyped green fluorescent protein (eGFP). **b** [AID]GANP cells were infected with influenza virus (WSN) expressing HA-pseudotyped eGFP protein and 2 h 45 min post-infection cells were treated with auxin to degrade the GANP protein. After 8 h of infection, cells were fixed and subjected to smRNA FISH to detect eGFP mRNA with probes labeled with Quasar 570. DNA was stained with Hoechst. Scale bar, 10 µm. Data are representative of three independent experiments. **c** Fluorescence intensity of eGFP mRNA was quantified in the whole cell and in the nucleus using the Imaris software (Bitplane). Percentage of nuclear values for each viral mRNA are shown for each cell (dot) (Untreated n = 34 cells and auxin treated n = 32 cells). Cells used for quantification were from three independent experiments. Data are presented as mean values +/− SD. p values were calculated using unpaired two tailed Student's t test (GraphPad Prism 9). p values ≤ 0.05 are considered significant.

## Viruses

All virus work was performed as per CDC guidelines for biosafety level 2. Influenza A viruses (A/WSN/33, A/Vietnam/1203/04, A/California/04/2009, and A/Wyoming/03/2003) were generated and tittered, as previously described[23]. Hemagglutinin-pseudotyped green fluorescent protein (eGFP)-expressing Influenza Virus was generated as described[24].

## Infection

Cells were washed with EMEM infection medium [EMEM (ATCC), 10 mM HEPES (Gibco), 0.125% BSA (Gibco), 0.5 µg/mL TPCK trypsin (Worthington Biomedical Corporation)] and incubated with A/WSN/33 for 1.5 h. Cells were then washed with EMEM infection medium and incubated in the same medium throughout the experiment. MOIs are indicated in the figures and/or legends.

## Plasmids

pCAG-EGFP was a gift from Wilson Wong (Addgene plasmid # 89684; http://n2t.net/addgene:89684; RRID: Addgene_89684)[25]. The 5' end of HA mRNA (nt 1-77) fused with eGFP and the 3' end of the HA mRNA (nt 1651-1754) were amplified from the pPOLI HA (45) GFP (80) vRNA plasmid[24] and cloned between MluI and NotI restriction sites to generate pCAG-HA5'end-eGFP-HA3'end, pCAG-HA5'end-eGFP, pCAG-eGFP-HA3'end and pCAG-HACDS(45nt)-eGFP plasmids. Single stranded oligos and eGFP PCR amplicons were assembled between MluI and NotI restriction sites to obtain pCAG-HA5'UTR-eGFP plasmids. All cloning was performed using the NEB HiFi assembly kit, according to the manufacturer's instructions.

## Transfections

For imaging experiments, cells were forward transfected with plasmids using lipofectamine 3000 (Invitrogen) in 24-well plates on glass coverslips (Fisher Scientific) coated with 0.2% gelatin (Sigma-Aldrich). For experiments using siRNAs (Dharmacon), A549 cells were reverse transfected with 30 nM siRNAs in 24-well plates using the Lipofectamine RNAiMAX Transfection Reagent (ThermoFisher), according to the manufacturer's instructions. Cells were incubated for 60 h and then infected with influenza viruses for 8 h. Cells were then subjected to smRNA FISH or western blot. Regarding the plaque assays shown in Fig. 3, cells were reverse transfected for 48 h and then infected as indicated in the figure legend.

## Minigenome assay

Cells were reverse transfected with equal quantities of pPOL1-HA(45) eGFP(80), pCDNA3 NP, pCDNA3 PA, pCDNA 3 PB1, pCDNA 3 PB2, pCAGGS NS1 and cultured on glass cover slips coated with 0.2% gelatin. After 24 h of transfection, cells were untreated or treated with auxin for 5 h 15 min and then subjected to smRNA FISH[24].

## smRNA FISH

smRNA FISH was performed as previously described[8]. M, NS and HA RNA FISH probes sequences are available in earlier articles[8,26]. The sequences of NP and eGFP mRNA probes are available in the Supplementary Tables 2 and 3.

## Fluorescence microscopy and data analysis

Image capturing and analysis were performed as previously described[8]. In brief, smRNA FISH images were captured using a Zeiss Axiovert 200 M automated microscope and the Micro-Manager version 2[27,28] or AxioVision 4.4 softwares. A Zeiss ×63 Plan-APOCHROMAT lens (1.4 numerical aperture) was used. Multiple Z planes of 0.3 µm thickness was captured to collect signals from both the cytoplasm and the nuclear compartments. Images were deconvolved using AutoQuant X v3.0.4 software. Deconvolved images were analyzed by Imaris 9.2 software (Bitplane) using surfaces tool for segmentation and signal analysis within the whole cell and the nucleus. Fiji software was used to process microscopy images to generate figures[29].

## Western blot

Cells were washed in PBS and harvested in 2× sample lysis buffer (125 mM Tris HCl pH 6.8, 20% glycerol, 4% SDS, 1× protease inhibitor cocktail). Lysate was subjected to sonication and 10 min boiling. Protein concentration was quantified using Bio-Rad DC Protein Assay Kit. Samples were loaded onto Bolt Bis-Tris Plus gels (Invitrogen) and subjected to western blot analysis as previously described[30]. Full scan blots can be found in the Source Data file.

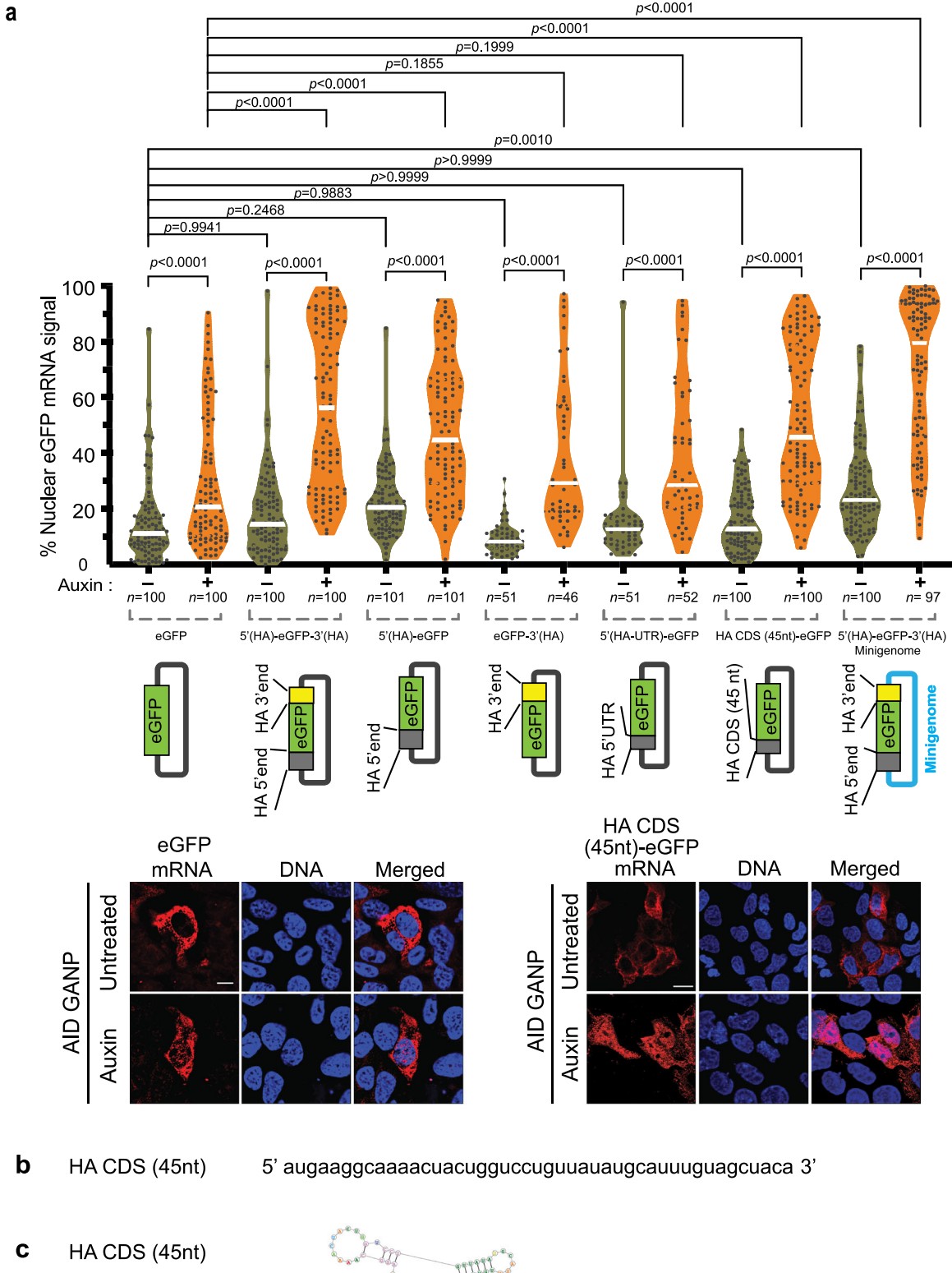

**b** HA CDS (45nt)     5' augaaggcaaaacuacugguccuguuauaugcauuuguagcuaca 3'

**c** HA CDS (45nt)
Energy= - 4.1 kcal/mol

**Extraction of cytoplasmic and nuclear RNA**

All samples and buffers were kept on ice throughout the experiment. All centrifugations were performed at 4 °C. Cells (100 mm dish) were washed in 10 ml ice cold PBS and then incubated in 9 ml of ice-cold PBS with 10 mM EDTA. After 3-4 min, cells were gently harvested using a scrapper and centrifuged for 5 min at 300 × $g$. Supernatant was removed and cells were gently washed with 5 ml of ice-cold PBS followed by centrifugation for 5 min at 300 × $g$. Supernatant was removed and gently suspended in ice-cold PBS to obtain 6 × 10^6 cells per ml, which were transferred to a pre-chilled 1.5 ml tube and centrifuged for 5 min at 300 × $g$. PBS was removed and cells were gently resuspend in 450 μl of buffer A (15 mM Tris-Cl pH 7.5, 15 mM NaCl, 60 mM KCl, 1 mM EDTA pH 8, 0.5 mM EGTA pH 8, 0.5 mM spermidine, 80U/ml RNAse inhibitor, 1× protease inhibitor cocktail). Buffer A (450 μl) with 0.4% NP-40 was then added to the cell suspension, which was mixed by inversion (6x), incubated for 5 min on ice, and centrifuged for 5 min at

**Fig. 7 | The 5′ end of the HA mRNA coding region encodes a 45 nt determinant for GANP- mediated mRNA export. a** Graph shows percentage of nuclear fluorescence in ᴬᴵᴰGANP cells transfected with pCAG plasmids encoding eGFP, HA5′end-eGFP-HA3′end, HA5′end-eGFP, eGFP-HA3′end, HA5′UTR-eGFP, and HA-CDS(45nt)-eGFP (the first 45 nt at the 5′ end of the HA coding sequence fused with eGFP) mRNAs, and with plasmids for the influenza virus minigenome assay expressing the HA5′end-eGFP-HA3′end mRNA. After 20 h of transfection, cells were untreated or treated with auxin for 5 h 15 min to degrade GANP. Cells were then subjected to smRNA FISH to detect eGFP mRNA with probes labeled with Quasar 570. DNA was stained with Hoechst. Quantification of fluorescence intensity was performed as in Fig. 6. "*n*" in the figure indicates the number of cells from three independent experiments used for quantification. *p* values were calculated using one-way ANOVA multiple comparisons (GraphPad Prism 9). *p* values ≤ 0.05 are considered significant. Representative images show ᴬᴵᴰGANP cells transfected with pCAG-eGFP plasmid or eGFP mRNA flanked by the first 45 nt at the 5′ end of the viral HA mRNA coding region (pCAG-HA-CDS(45nt)-eGFP). Scale bar, 10 μm. Data are representative of three independent experiments. **b** The first 45 nt of the coding region of the HA mRNA are shown. **c** Predicted structure of the HA sequence in **b** was determined using the RNAStructure software.

400 × *g*. Supernatant (700 μl) was removed without disturbing the nuclear pellet and transferred to a new tube on ice (cytoplasmic fraction). The nuclear pellet was gently suspended in 450 μl RLN buffer (50 mM Tris-Cl pH 7.5, 140 mM NaCl, 1.5 mM MgCl₂, 10 mM EDTA pH 8, 80 U/ml RNAse inhibitor, 1× protease inhibitor cocktail). RLN buffer (450 μl) with 0.5% NP40 was added to the nuclear pellet, which was then mixed by inversion, and incubated for 3 min on ice. While nuclear pellet was incubated on ice, the cytoplasmic fraction was centrifuged for 1 min at 500 × *g* and 550 μl was transferred into a new tube on ice. The nuclear fraction was centrifuged for 5 min at 500 × *g* immediately after the 3 min incubation. Supernatant was removed from the nuclear pellet and the nuclear pellet was placed on ice. The cytoplasmic fraction was centrifuged at 9300 × *g* for 2 min and 450 μl of the supernatant was transferred to a new tube placed on ice (final cytoplasmic fraction) and then mixed with 3 times volume of LS-TRIzol (Thermo Fisher). The nuclear pellet (final nuclear fraction) was suspended in 1 ml regular TRIzol. Pipetted up and down to completely resuspend the nuclear pellet. RNA was isolated according to the manufacturer's instructions. Finally, RNA was dissolved in 100 μl nuclease-free water. RNA was further purified using the Qiagen RNeasy plus kit with the modifications below. The RLT plus buffer (350 μl) was added to 100 μl RNA, mixed well, and passed through gDNA columns according to the manufacturer's instructions. Ethanol (100%; 250 μl) was added instead of 350 μl of 70% ethanol. Subsequent steps were performed as per manufacturer's instructions. To increase RNA yield, pass RNA elutes again through the same column in the last step. This is a modified version of a fractionation protocol from Igor Ulitsky's laboratory.

### RNA-seq and data analysis
RNA samples were subjected to RNA-Seq as previously described[26]. RNA-Seq was performed with samples from two independent experiments. In brief, RNA quality was determined using the Agilent 2100 Bioanalyzer (RIN Score 8 or higher). The Qubit fluorometer was used for determining RNA concentration. Strand-specific cDNA synthesis was performed after poly(A) RNA was purified and fragmented. Samples were sequenced on the Illumina HiSeq 2500 with paired end 150 bp reads. Raw sequence data was trimmed using Trimmomatic[31]. QC filtered trimmed sequences were aligned to hg19 using STAR[32]. All subsequent analysis was performed using R version 4.0.2 and Bioconductor 3.11 in RStudio[33]. Raw counts were obtained from BAM files using FeatureCounts from the Rsubread package[34], and TPM was calculated from raw counts. The DESeq2 package was used for differential expression analysis between control and auxin treated cells, with Benjamini and Hochberg correction (FDR < 0.05)[35]. Transcripts whose levels were not significantly altered ($1 < \log_2$ fold change > −1) in the total cell lysate of GANP-depleted cells versus control cells were considered for further analysis to identify export inhibited and not affected mRNAs. TPM values of transcripts from nuclear fraction and cytoplasmic fraction were used to calculate N/C ratio for untreated and GANP depleted samples. Relative change in N/C ratio was calculated comparing N/C ratio of GANP depleted sample with untreated sample. Transcripts with relative change in N/C ratio >3.5 in two independent experiments were considered as export inhibited mRNAs and transcripts with relative change in N/C ratio between 1.5 and 0.66 in two

independent experiments were considered as not affected mRNAs. Features of export inhibited and not affected transcripts were extracted based on Ensembl GTF annotations (GRCh38, release version 104) for each Ensembl Canonical transcript. These features represent information regarding the number, length, and GC content of the exons and introns of each transcript as well the entire transcript or various mRNA transcript regions (i.e., the 5′UTR, CDS, and 3′UTR). Boxplots were plotted to compare the distribution of these feature values between the export inhibited and not affected groups and significance was assessed using a two-tailed Mann–Whitney *U*-rank test in Python.

### qRT-PCR
SuperScript™ II Reverse Transcriptase (Invitrogen) was used to synthesize cDNA using random hexamers (for cellular mRNAs) or oligo d(T) and 18 S reverse primer mix (for viral mRNAs). cDNA (1:3 diluted) was mixed with Roche 480 SYBR Green I Master, as per manufacturer's instructions, and primers were added to the samples in qRT-PCR 96-well plates. Quantitative real-time PCR was performed using the Roche LightCycler 480. Primer sequences are provided in Supplementary Table 4.

### BV-BRC database and NCBI database sequence alignment
The uridines in the 45-nucleotide HA mRNA sequence were first replaced by thymidine. The sequence was then blasted in the BV-BRC database (https://www.bv-brc.org/) with alignment parameters specifying the search to be within a taxon of order Articulavirales with the database type set to "genes" and with the maximum hits parameter set to a 100. The NCBI database (https://blast.ncbi.nlm.nih.gov/Blast.cgi?PROGRAM = blastn&PAGE_TYPE = BlastSearch&LINK_LOC = blasthome) was also used to blast the sequence using the default setting.

### RNA structural predictions
The mRNA sequences were inputted in the RNAstructure webserver under "predict a secondary structure" using default parameters. The Connectivity Table (CT) file is then extracted and inputted into VARNA for optimum visualization quality.

### Reporting summary
Further information on research design is available in the Nature Portfolio Reporting Summary linked to this article.

## Data availability
The RNA-Seq data are deposited in the NCBI Sequence Read Archive (SRA) with the BioProject accession code PRJNA882571. The proteomics data on the characterization of PCID2 cells have been deposited to the ProteomeXchange Consortium via the PRIDE partner repository with the dataset identifier PXD039407. There are several files with raw data and a summary file titled "Results_partial.xls." Source data are provided with this paper.

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

## Acknowledgements

This work was supported by NIH R01 AI154635 to B.M.A.F.; NIH R01 AI125524 to K.W.L. and B.M.A.F.; this work was also partly supported by CRIPT (Center for Research of Influenza Pathogenesis and Transmission), an NIAID Center of Excellence for Influenza Research and Response (CEIRR, contract number 75N93021C00014) and NIAID grant U19AI135972 to A.G.-S. V.A. and M.D. were supported by the Intramural Research Program of the *Eunice Kennedy Shriver* National Institutes of Child Health and Human Development at NIH (Intramural Project # Z01 HD008954). B.M.A.F. holds the Ruth S. Harrell Professorship in Medical Research.

## Author contributions

P.B. and B.M.A.F. designed the project, analyzed the data, and wrote the paper. P.B., V.A., E.A.R., S.A., S.G., M.E., T.C., K.B., S.S.E.Z., and A.A. performed experiments and analyzed data. H.Z., V.A., A.G.-S., and M.D. developed reagents. M.D., B.S.T., K.W.L., A.G.-S., and J.W.S. analyzed data. K.B., M.G., and C.H. performed bioinformatics analysis. P.B., B.M.A.F., V.A., J.W.S., H.Z., K.W.L., T.C., and A.G.-S. edited the manuscript.

## Competing interests

The authors declare no competing interests. The A.G.-S. laboratory has received research support from Pfizer, Senhwa Biosciences, Kenall Manufacturing, Blade Therapeutics, Avimex, Johnson & Johnson, Dynavax, 7Hills Pharma, Pharmamar, ImmunityBio, Accurius, Nanocomposix, Hexamer, N-fold LLC, Model Medicines, Atea Pharma, Applied Biological Laboratories and Merck, outside of the reported work. A.G.-S. has consulting agreements for the following companies involving cash and/or stock: Castlevax, Amovir, Vivaldi Biosciences, Contrafect, 7Hills Pharma, Avimex, Vaxalto, Pagoda, Accurius, Esperovax, Farmak, Applied Biological Laboratories, Pharmamar, Paratus, CureLab Oncology, CureLab Veterinary, Synairgen and Pfizer, outside of the reported work. A.G.-S. has been an invited speaker in meeting events organized by Seqirus, Janssen, Abbott and Astrazeneca. A.G.-S. is inventor on patents and patent applications on the use of antivirals and vaccines for the

treatment and prevention of virus infections and cancer, owned by the Icahn School of Medicine at Mount Sinai, New York, outside of the reported work.
