## [Peer Review File · Nature Communications]

Reviewers' Comments:

Reviewer #1:

Remarks to the Author:

In the manuscript by Bhat et al (NCOMMS-22-36268) the authors explored the components of the mRNA export pathway that mediates IAV mRNA export through the nuclear pore complex. By using the Auxin-Induced Degron (AID) system to degrade proteins of the TREX-2 complex they show that this complex is required for efficient IAV mRNA export. Furthermore, a nucleotide stretch in the HA mRNA was identified to be sufficient for interaction with the complex.

This is an interesting work that sheds light on the specific regulation of export of IAV mRNA. Some points, however, still have loose ends regarding the general applicability of the concept and need to be addressed.

Major:

Figure 1 and 2: Why there is such a pronounced decrease of AID-TPR already in non-treated cells? Is this a common observation with AID-tagged proteins? Does this point to a general deregulation or instability of AID-tagged proteins? This should be discussed and the respective untreated controls should also be done in Figure 2 with AID-GANP and AID-PCID2.

Do the M and NS probes only detect unspliced mRNA or both, spliced and unspliced mRNA? The information on the sequences is somehow hidden also in the cited literature and a clear statement on that is required. If probes to unspliced mRNA of M and NS are used, it would also be required to do a test with probes to spliced mRNA for a proof of the general validity of the concept.

DLD-1 cells are not commonly used experimentally as host cells for IAV. As a human colon cancer cell line that expresses a truncated adenomatous polyposis coli (APC) protein, cellular regulation might be messed up which could also effect mRNA export.

While the results are clean for M, NS and HA (and NP) mRNAs it would be desirable to see effects on the whole set of viral mRNAs from each segment.

A/WSN/33 is a laboratory strain and would not necessarily reflect the situation of all IAV. Some of the key data should be repeated with other more natural IAV isolates.

While the authors convincingly show the importance of the 45 nucleotide region in HA mRNA it would be desirable to also identify similar regions in other viral mRNAs, again for a proof of the general concept.

Minor:

Please introduce all abbreviations in the introduction and the main body of the manuscript.

Reviewer #2:

Remarks to the Author:

In the manuscript, Bhat and colleagues demonstrate that TPR and components of TREX2 are required for the efficient nuclear export of influenza mRNAs, which are intronless or intron-poor. Overall, the manuscript is well written and presents results that are of great interest to the mRNA biology and virology communities. I only have a few comments that should be addressed.

- It is unclear whether TPR, GANP and PCID2 are required for the export of both spliced and unspliced forms of the M and NS mRNAs. The authors should try to address this.
- It is interesting that those mRNAs that are TPR/TREX2-dependent have relatively low GC-content. This is in agreement with what was seen in Zuckerman Mol Cell 2020. Despite this, Zuckerman report that GC-rich motifs near the 5' end of the transcripts ultimately promote TPR-dependent export. In our analysis of mRNAs that are TPR-dependent (Lee NAR 2020), we find that

these mRNAs do tend to have elevated GC-content at their 5'end (Lee & Palazzo, unpublished data). In light of this, I think it would be helpful if the authors reanalyzed their data, but looking at GC-content at the 5'end – typically this region is within the first 200 nucleotides from the transcription start site (see Palazzo & Kang Bioessays 2021).

- I could not find the analysis of Malat1 and GAPDH mRNA levels in the nuclear and cytosolic fractions to analyze the relative purity of the fractionation protocol. These should be provided.

- In the cytoplasmic/nuclear fractionations was the ER effectively solubilized? This is a problem with some protocols and can skew the results as mRNAs that are bound to the ER could co-segregate with the nucleus. Analyzing a well-expressed ER-associated mRNA, such as calreticulin or BiP, should address this.

- The sequencing data should be deposited in an online repository (e.g. GEO). I could not find this information in the manuscript or supplementary file.

In general I am supportive of publication, provided that these concerns are addressed.

Sincerely,
Alex Palazzo

Reviewer #3:

Remarks to the Author:

Bhat et al., present a compelling case for the export of some unspliced IAV mRNAs from the nucleus to the cytoplasm via TREX-2 complex, mainly through manipulation of GANP expression in cell culture. This is a very pertinent question in the virology field, that can also have wider implication for the trafficking of cellular mRNAs, as shown by the authors in Figure 4. Through the use of an Auxin-induced degron system the authors manipulate the expression of components of the TREX-2 complex to show that their expression correlates with M, NS and HA mRNA translocating from the nucleus to the cytoplasm.

I however have a number of concerns regarding the methodology that I would like to see rectified before publication.

1. These conclusion rely heavily on quantification of smFISH images. It would be worthwhile to also perform nuclear:cytoplasmic fractionation of infected + induced cells followed by qPCR to determine that the smFISH results hold up by another method. This should be relatively straightforward for the authors to do as they have already demonstrated the capabilities to perform fractionation experiments followed by qPCR in Figure S4.

2. I would expect that inhibiting the export of IAV mRNAs to such a great extent as is throughout the manuscript would lead to gross viral phenotypes such as depletion of viral proteins and reduction in viral titres, which can easily be checked by Western blot and plaque assay, respectively. These experiments should be performed as it would point to the necessity of the TREX-2 complex for overall IAV replication.

3. Though I understand the authors are looking for the exact RNA element that may lead to recruitment of the TREX-2 complex, I find Figure 5 quite noisy and not at all convincing. I'm not sure if this is due to the quantification method itself or the use of the AID system as the addition of Auxin alone seems to have an impact on GFP trafficking with the average before Auxin appearing to be around 10% nuclear while after the addition of auxin it rises to approximately 35-40%. Also, the addition of the 45nt CDS seems to have a positive impact on nuclear retention of eGFP mRNA in both auxin – and + samples. Unfortunately, the authors have only calculated significance within the auxin treated samples. They should also calculate whether the differences in Auxin – and + samples are significant for each plasmid. Perhaps if this small 45nt element could be tested on the 5'UTR of additional reporter genes that would serve to better convince readers. I understand that this would require buying more sets of smFISH probes which is very expensive, and I don't believe very fair to the authors. But I think just repeating the experiment with some additional reporter genes and performing nuc:cyto fractionation and qPCR would represent a more convincing argument.

4. As the authors only look at the WSN strain and only look at the trafficking of 4 viral mRNAs in I think the title and abstract is misleading as this has not been proven as a mechanism for influenza A mRNA export. If this title and abstract are going to remain in place then multiple IAV strains should be tested to confirm that this mechanism of mRNA export is conserved among many IAVs.

5. Could the author comment on the conservation of the HA 45nt section that they speculate assists with TREX-2 mediated export? I think it would be beneficial to at least include some analysis within the supplemental of this using the thousands of influenza A sequences available in public databases. As an aside, this could also hint at smaller greatly conserved regions within the 45nt section that might be particularly interesting to investigate further by mutagenesis, but those experiments would be beyond this current work of course. But the conservation analysis would be very beneficial.

Overall, I think this work is very important and extremely interesting for virologists and cell biologists and if the above mentioned experiments are performed, which I don't think should be overly time-consuming to the authors, then this study will solidify a role for TREX-2 in the export of some WSN specific mRNAs.

Response To Reviewers

We thank all Reviewers for the constructive comments. Below is a point-by-point response to the comments, including additional data added to the manuscript, as requested.

Reviewer #1 (Remarks to the Author):

"In the manuscript by Bhat et al (NCOMMS-22-36268) the authors explored the components of the mRNA export pathway that mediates IAV mRNA export through the nuclear pore complex. By using the Auxin-Induced Degron (AID) system to degrade proteins of the TREX-2 complex they show that this complex is required for efficient IAV mRNA export. Furthermore, a nucleotide stretch in the HA mRNA was identified to be sufficient for interaction with the complex."

"This is an interesting work that sheds light on the specific regulation of export of IAV mRNA. Some points, however, still have loose ends regarding the general applicability of the concept and need to be addressed."

"Major:"

"Figure 1 and 2: Why there is such a pronounced decrease of AID-TPR already in non-treated cells? Is this a common observation with AID-tagged proteins? Does this point to a general deregulation or instability of AID-tagged proteins? This should be discussed and the respective untreated controls should also be done in Figure 2 with AID-GANP and AID-PCID2."

Response: Leaky degradation of certain AID-tagged proteins can occur, as previously reported (PMID: 31026591; PMID: 31451765; PMID: 31467088). This is the case for the ^{AID}TPR that showed inhibition of viral mRNA export due to decreased TPR levels. Therefore, we added the depicted parental DLD-1 cell as control. In the case of ^{AID}GANP and ^{AID}PCID2 cells, the nucleocytoplasmic distribution of the viral M, NS and HA mRNAs in the absence of auxin (Fig. 2 images and graphs) was not altered as compared to the nucleocytoplasmic distribution of these viral mRNAs in DLD-1 parental cell line (Fig. 1p-u). Thus, the DLD-1 parental cell line is the same for all AID-tagged proteins. This has been clarified in the text (page 5, lines 9-10). Additionally, this result is further confirmed by subcellular fractionation in which additional viral mRNAs were quantified in nuclear and cytoplasmic fractions and are shown to be significantly retained in the nucleus upon Tpr, GANP, or PCID2 degradation (see new Supplementary Fig. 1). Moreover, we further validated this finding using siRNAs in A549 cells (see new Supplementary Fig. 3c-k).

"Do the M and NS probes only detect unspliced mRNA or both, spliced and unspliced mRNA? The information on the sequences is somehow hidden also in the cited literature and a clear statement on that is required. If probes to unspliced mRNA of M and NS are used, it would also be required to do a test with probes to spliced mRNA for a proof of the general validity of the concept."

Response: The M and NS probes detect both unspliced and spliced forms. This has been clarified in the text (page 4, lines 14-15). Additionally, as requested by another reviewer and mentioned above, we

have also quantified additional viral mRNAs by qPCR in nuclear and cytoplasmic fractions of TPR, GANP and PCID2 depleted cells versus their respective controls (new Supplementary Fig. 1).

“DLD-1 cells are not commonly used experimentally as host cells for IAV. As a human colon cancer cell line that expresses a truncated adenomatous polyposis coli (APC) protein, cellular regulation might be messed up which could also affect mRNA export.”

Response: We have repeated the same smRNA-FISH experiments in A549 cells (control versus GANP knockdown by siRNAs) to detect viral mRNAs. The results are the same as we observed upon GANP degradation by auxin in DLD-1 cells. Knockdown of GANP in A549 cells inhibited viral mRNA export (see new Supplementary Fig. 3c-k). We have also performed viral replication studies and measured viral protein levels in A549 cells upon GANP knockdown. As expected, a decrease in viral protein levels and inhibition of viral replication were observed upon GANP knockdown (see new Fig. 3).

“While the results are clean for M, NS and HA (and NP) mRNAs it would be desirable to see effects on the whole set of viral mRNAs from each segment.”

Response: As mentioned above, we performed subcellular fractionation in which additional viral mRNAs were quantified in nuclear and cytoplasmic fractions and are shown to be significantly retained in the nucleus upon GANP, TPR, or PCID2 degradation (see new Supplementary Fig. 1).

“A/WSN/33 is a laboratory strain and would not necessarily reflect the situation of all IAV. Some of the key data should be repeated with other more natural IAV isolates.”

Response: As requested, we have repeated the key data with diverse strains (A/California/07/2009; A/Wyoming/03/2003; A/Vietnam/2506/2004) and added viral protein levels and viral replication studies, as mentioned above. Please see Fig. 3 and Supplementary Fig. 3.

“While the authors convincingly show the importance of the 45 nucleotide region in HA mRNA it would be desirable to also identify similar regions in other viral mRNAs, again for a proof of the general concept.”

Response: We compared this region with other viral mRNAs. The bioinformatic findings showed that it is possible to have a combination of sequence and stem-loop structures in the 5' end of the viral mRNAs that could determine the signal. However, these should be extensively validated experimentally with diverse mutations and RNA structural studies. Therefore, we feel that this is beyond the scope of the current manuscript, and we will be investigating these features in follow-up studies. Instead, as we described in the results section, we aligned the 45-nucleotide HA mRNA sequence against the BV-BRC database and the NCBI database to record sequence conservation. Similar sequence alignment results were obtained in both databases. The 45-nucleotide region of HA mRNAs was shown to be conserved across several influenza A strains. The first 13 strains depicted in Supplementary Fig. 9a are shown to be identical, while strains numbered 13 through 22 have shown to hold at least a 91% identity match. In addition to sequence conservation, secondary structure predictions were compared between these 22 strains using the *RNAstructure* webserver. The secondary structures of the different sequences obtained from the sequence alignment were predicted and grouped in Supplementary Fig. 9b according to their sequence similarities. The first secondary structure corresponding to strains 1-13, which are identical to the 45-nucleotide HA mRNA region, has shown to form two distinct stem-loops. The first demonstrating

a low probability of folding of around 50%, while the second stem-loop shows a higher probability of 80%. In strain 14, the single nucleotide change caused the first stem-loop to unfold, but maintains the formation of the second stem-loop, which is identical to the original structure. In strains 15 through 20, a three-nucleotide alteration caused a complete alteration in the predicted secondary structure compared to the HA mRNA region. Similar observations can be made for strains 21 and 22 with a folding probability of 50%. These low probabilities can be attributed to their dependence on other stem-loops in close proximity to stabilize their structures. These findings layout the foundation for future projects.

“Minor:”

“Please introduce all abbreviations in the introduction and the main body of the manuscript.”

Response: We addressed this point as requested. We added an Abbreviations section in the first page of the manuscript and also included the description of the abbreviations in the main body of the text (Introduction section).

Reviewer #2 (Remarks to the Author):

“In the manuscript, Bhat and colleagues demonstrate that TPR and components of TREX2 are required for the efficient nuclear export of influenza mRNAs, which are intronless or intron-poor. Overall, the manuscript is well written and presents results that are of great interest to the mRNA biology and virology communities. I only have a few comments that should be addressed.”

Response: We thank the Reviewer for the constructive comments.

“- It is unclear whether TPR, GANP and PCID2 are required for the export of both spliced and unspliced forms of the M and NS mRNAs. The authors should try to address this.”

Response: As requested, we added new data showing that TPR, GANP, and PCID2 are also required for nuclear export of the spliced forms of M and NS mRNAs as well as for additional viral mRNAs not previously tested. These were performed by subcellular fractionation in which the levels of viral mRNAs were measured in nuclear and cytoplasmic fractions by RT-qPCR (see new Supplementary Fig.1).

“- It is interesting that those mRNAs that are TPR/TREX2-dependent have relatively low GC-content. This is in agreement with what was seen in Zuckerman Mol Cell 2020. Despite this, Zuckerman report that GC-rich motifs near the 5’end of the transcripts ultimately promote TPR-dependent export. In our analysis of mRNAs that are TPR-dependent (Lee NAR 2020), we find that these mRNAs do tend to have elevated GC-content at their 5’end (Lee & Palazzo, unpublished data). In light of this, I think it would be helpful if the authors reanalyzed their data, but looking at GC-content at the 5’end – typically this region is within the first 200 nucleotides from the transcription start site (see Palazzo & Kang Bioessays 2021).”

Response: As requested, we have analyzed the GC content of the 5’end. While we did not find a significant difference between the two gene sets when looking at the first 200 nt, we found significant difference when analyzing the first 75 nt or 45 nt in which the mRNAs not affected by GANP degradation show higher GC% than mRNAs whose export are GANP-dependent. Please see new Supplementary Fig. 6.

“- I could not find the analysis of Malat1 and GAPDH mRNA levels in the nuclear and cytosolic fractions to analyze the relative purity of the fractionation protocol. These should be provided.”

Response: Please see Supplementary Fig. 5.

“- In the cytoplasmic/nuclear fractionations was the ER effectively solubilized? This is a problem with some protocols and can skew the results as mRNAs that are bound to the ER could co-segregate with the nucleus. Analyzing a well-expressed ER-associated mRNA, such as calreticulin or BiP, should address this.”

Response: Calreticulin is shown in Supplementary Fig. 5b.

“- The sequencing data should be deposited in an online repository (e.g. GEO). I could not find this information in the manuscript or supplementary file.”

Response: This information was submitted as part of the Reporting Summary. The RNA-Seq data is deposited in the NCBI Sequence Read Archive (SRA) (<https://dataview.ncbi.nlm.nih.gov/object/PRJNA882571?reviewer=u6i7hcaw8rodi1lnlovaghmkp3>) with the BioProject accession code PRJNA882571.

This information is now added in the Data Availability Section of the main text.

“In general I am supportive of publication, provided that these concerns are addressed.”

*Sincerely,
Alex Palazzo*

Reviewer #3 (Remarks to the Author):

“Bhat et al., present a compelling case for the export of some unspliced IAV mRNAs from the nucleus to the cytoplasm via TREX-2 complex, mainly through manipulation of GANP expression in cell culture. This is a very pertinent question in the virology field, that can also have wider implication for the trafficking of cellular mRNAs, as shown by the authors in Figure 4. Through the use of an Auxin-induced degron system the authors manipulate the expression of components of the TREX-2 complex to show that their expression correlates with M, NS and HA mRNA translocating from the nucleus to the cytoplasm.”

“I however have a number of concerns regarding the methodology that I would like to see rectified before publication.”

“1. These conclusion rely heavily on quantification of smFISH images. It would be worthwhile to also perform nuclear:cytoplasmic fractionation of infected + induced cells followed by qPCR to determine that the smFISH results hold up by another method. This should be relatively straightforward for the authors to do as they have already demonstrated the capabilities to perform fractionation experiments followed by qPCR in Figure S4.”

Response: As requested, we performed nuclear:cytoplasmic fractionation and further confirmed that influenza virus mRNAs are retained in the nucleus upon GANP, TPR, or PCID2 degradation (see new Supplementary Fig. 1).

“2. I would expect that inhibiting the export of IAV mRNAs to such a great extent as is throughout the manuscript would lead to gross viral phenotypes such as depletion of viral proteins and reduction in viral titres, which can easily be checked by Western blot and plaque assay, respectively. These experiments should be performed as it would point to the necessity of the TREX-2 complex for overall IAV replication.”

Response: Yes, decreasing nuclear export of viral mRNAs did decrease viral protein levels and viral replication. Please see new data in Fig. 3.

“3. Though I understand the authors are looking for the exact RNA element that may lead to recruitment of the TREX-2 complex, I find Figure 5 quite noisy and not at all convincing. I’m not sure if this is due to the quantification method itself or the use of the AID system as the addition of Auxin alone seems to have an impact on GFP trafficking with the average before Auxin appearing to be around 10% nuclear while after the addition of auxin it rises to approximately 35-40%. Also, the addition of the 45nt CDS seems to have a positive impact on nuclear retention of eGFP mRNA in both auxin – and + samples. Unfortunately, the authors have only calculated significance within the auxin treated samples. They should also calculate whether the differences in Auxin – and + samples are significant for each plasmid. Perhaps if this small 45nt element could be tested on the 5’UTR of additional reporter genes that would serve to better convince readers. I understand that this would require buying more sets of smFISH probes which is very expensive, and I don’t believe very fair to the authors. But I think just repeating the experiment with some additional reporter genes and performing nuc:cyto fractionation and qPCR would represent a more convincing argument.”

Response: As requested, we now also show the p values for the data without auxin compared to + auxin. As mentioned in the text, nuclear export of eGFP mRNA shows low GANP-dependency. To further improve data quantification, we repeated the experiments and analyzed ~100 cells per construct. The cell heterogeneity we obtain here is similar to what others observe in the field, such as in Zuckerman B. et al, *Mol. Cell* 2020. We have also tried to use luciferase as reporter but due to its low GC content, we found that it is a GANP-dependent mRNA, therefore it was not useful for this purpose. β -galactosidase has GC content between eGFP and Luciferase, therefore it is not useful either. The ideal would be to develop constructs that would further reflect the GANP rules for export based on the knowledge of viral and cellular mRNAs, but these would constitute future studies. To further validate this finding, we used the viral minigenome system, in which the 5’ and 3’ ends of the viral HA mRNA flanking eGFP was expressed in the presence of the viral polymerase. In this case, the results showed a robust and more homogeneous GANP-dependency compared to the expression of the same mRNA in the context of plasmid. This further corroborates the phenotype. These data is now in Figure 7.

“4. As the authors only look at the WSN strain and only look at the trafficking of 4 viral mRNAs in I think the title and abstract is misleading as this has not been proven as a mechanism for influenza A mRNA export. If this title and abstract are going to remain in place then multiple IAV strains should be tested to confirm that this mechanism of mRNA export is conserved among many IAVs.”

Response: As requested, we tested additional influenza viruses and the results show that the mRNA export mechanism is conserved among other strains (see new Supplementary Fig. 3). Additionally, we show that viral protein levels and virus replication are also consequently reduced upon GANP knockdown (please see new Fig. 3).

“5. Could the author comment on the conservation of the HA 45nt section that they speculate assists with TREX-2 mediated export? I think it would be beneficial to at least include some analysis within the supplemental of this using the thousands of influenza A sequences available in public databases. As an aside, this could also hint at smaller greatly conserved regions within the 45nt section that might be particularly interesting to investigate further by mutagenesis, but those experiments would be beyond this current work of course. But the conservation analysis would be very beneficial.”

Response: We aligned the 45-nucleotide HA mRNA sequence against the BV-BRC database and the NCBI database to record sequence conservation. Similar sequence alignment results were obtained in both databases. The 45-nucleotide region of HA mRNAs was shown to be conserved across several influenza A strains. The first 13 strains depicted in Supplementary Fig. 9a are shown to be identical, while strains numbered 13 through 22 have shown to hold at least a 91% identity match. In addition to sequence conservation, secondary structure predictions were compared between these 22 strains using the *RNAstructure* webserver. The secondary structures of the different sequences obtained from the sequence alignment were predicted and grouped in Supplementary Fig. 9b according to their sequence similarities. The first secondary structure corresponding to strains 1-13, which are identical to the 45-nucleotide HA mRNA region, has shown to form two distinct stem-loops. The first demonstrating a low probability of folding of around 50%, while the second stem-loop shows a higher probability of 80%. In strain 14, the single nucleotide change caused the first stem-loop to unfold, but maintains the formation of the second stem-loop, which is identical to the original structure. In strains 15 through 20, a three-nucleotide alteration caused a complete alteration in the predicted secondary structure compared to the HA mRNA region. Similar observations can be made for strains 21 and 22 with a folding probability of 50%. These low probabilities can be attributed to their dependence on other stem-loops in close proximity to stabilize their structures. These findings lay out the foundation for future projects.

“Overall, I think this work is very important and extremely interesting for virologists and cell biologists and if the above mentioned experiments are performed, which I don't think should be overly time-consuming to the authors, then this study will solidify a role for TREX-2 in the export of some WSN specific mRNAs.”

Response: Thank you.

Reviewers' Comments:

Reviewer #1:

Remarks to the Author:

The authors have convincingly addressed most of my concerns. While it would be still desirable to expand the study on the 45nt stretch in the HA to other gene segments, I agree with the authors that this would probably exceed the scope of the current manuscript.

Reviewer #2:

Remarks to the Author:

The authors have addressed all my concerns.

Reviewer #3:

Remarks to the Author:

All of my comments have been adequately addressed by the authors and I am happy for this paper to now be accepted for publication.

Response To Reviewers

Reviewers have not asked for additional experiments. Please see below.

“REVIEWERS' COMMENTS

Reviewer #1 (Remarks to the Author):

The authors have convincingly addressed most of my concerns. While it would be still desirable to expand the study on the 45nt stretch in the HA to other gene segments, I agree with the authors that this would probably exceed the scope of the current manuscript.

Reviewer #2 (Remarks to the Author):

The authors have addressed all my concerns.

Reviewer #3 (Remarks to the Author):

All of my comments have been adequately addressed by the authors and I am happy for this paper to now be accepted for publication.”